RESEARCH ARTICLE

# Is parity a cause of tooth loss? Perceptions of northern Nigerian Hausa women

**Elizabeth O. Oziegbe[1,2], Lynne A. Schepartz[2,3]** *

**1** Faculty of Dentistry, Obafemi Awolowo University, Ile-Ife, Nigeria, **2** Human Variation and Identification Research Unit (HVIRU), School of Anatomical Sciences, Faculty of Health Sciences, University of the Witwatersrand, Johannesburg, South Africa, **3** University of Pennsylvania Museum of Archaeology and Anthropology, Philadelphia, Pennsylvania, United States of America

* Lynne.Schepartz@wits.ac.za

OPEN ACCESS

**Data Availability Statement:** All relevant data are within the manuscript and its Supporting Information files.

**Funding:** LAS used funding received from the National Research Foundation of South Africa as a

## Abstract

### Background

Reproduction affects the general health of women, especially when parity is high. The relationship between parity and oral health is not as clear, although it is a widespread customary belief that pregnancy results in tooth loss. Parity has been associated with tooth loss in some populations, but not in others. It is important to understand the perceptions of women regarding the association between parity and tooth loss as these beliefs may influence health behaviors during the reproductive years.

### Aim

To explore the views of Hausa women regarding the link between parity and tooth loss.

### Methods

Qualitative data were collected through a grounded theory approach with focus group discussions (FGDs) of high and low parity Hausa women (n = 33) in northern Nigeria. Responses were elicited on the causes of tooth loss, effects of tooth loss on women's quality of life, issues of parity and tooth loss, and cultural beliefs about parity and tooth loss. The data were analyzed thematically using ATLAS-ti.

### Results

Respondents associated tooth loss with vomiting during labor, a condition termed 'payar baka'. Poor oral hygiene, excessive consumption of refined carbohydrates, tooth worm, cancer and ageing were also believed to cause tooth loss. The greatest impacts of tooth loss on the lives of the respondents were esthetic and masticatory changes.

### Conclusion

Respondents perceived that parity is indirectly linked to tooth loss, as reflected in their views on the association between vomiting during labor and tooth loss.

rated research to support this study. The funders had no role in study design, data collection and analysis, decision to publish, or preparation of the manuscript.

**Competing interests:** The authors have declared that no competing interests exist.

## Introduction

"A tooth for every child" is a common phrase in many societies [1]. Women frequently trace their oral health problems to their reproductive years. A woman looks forward to being a mother with great joy, but could it be at a cost to her oral health? Pregnancy, breastfeeding and childcare are associated with physiological, metabolic and energetic demands. The stresses of repeated pregnancies can lead to permanent changes in health, especially when parity is high [2]. These changes can include negative effects on oral health. It is posited that the cumulative effects of nutritional stress from repeated pregnancies cause calcium depletion from bones and tissues in the oral cavity, with subsequent loss of teeth [3]. The notion of pregnancy draining calcium directly from teeth is often raised, but it has not been substantiated [4–6].

Parity research has focused on the effects of parity on general health, with few studies on oral health. General associations between higher parity and more teeth lost are documented among women of European ancestry [1,7–9]. There are few data for African women or women in rural settings. No study has focused on how women view the role of parity in tooth loss. Consequently, this study addresses a gap in the literature concerning information on the perceptions of African women regarding parity and tooth loss.

### Tooth loss

In the past, tooth loss was considered part of normal aging and unavoidable [10]. Presently, the reverse is the case, particularly in areas where dental care is available. The loss of one or more teeth can affect an individual's quality of life [11]. Aside from obvious changes in appearance and mastication, tooth loss can affect self-esteem, speech and social interaction. Koyama *et al.* [12] found that higher levels of social engagement were associated with less tooth loss in older Japanese individuals.

Qualitative and quantitative studies on individuals with complete or partial tooth loss document serious negative emotional consequences [13–15]. Completely edentulous patients reported bereavement, loss of self-confidence, concerns about appearance and self-image, tooth loss as a taboo subject that could not be discussed with other people, and keeping tooth loss secret [14]. Partially dentate patients described tooth loss as a sense of being incomplete and having lost a part of their body [13].

There are various causes of tooth loss. The principal contributors are caries and periodontal disease; others include trauma, orthodontic treatment, tooth impaction, cystic lesions and neoplasms [10,16]. Cultural identity markers also incorporate evulsion of the highly visible incisors and canines [17]. Globally, caries is the principal reason for tooth loss along with extraction following periodontal disease. This characterizes countries with extensive dental care facilities as well as developing countries [18–21].

Age, gender, socio-economic status, level of education and utilization of oral health services are associated with differing levels of tooth loss [12,22,23]. Individuals with low socio-economic status overwhelmingly have the worst oral health [24,25], yet lower level of literacy and poor attitudes towards dental check-ups also are associated with tooth loss [26,27].

Gender-based dietary differences, eating patterns, resource availability and cultural attitudes towards health, pain and dentistry underlie male-female differences in tooth loss [28,29]. Tooth loss frequency varies by sex in most societies and appears to be context dependent. Some studies describe sex as a significant predictor of tooth loss with a tendency towards higher prevalences in females [28,30–33], while others report higher levels in males [34,35].

The observed male-female differences in tooth loss are usually attributed to changes in female sex hormones during pregnancy, with a logical inference that parity is related to tooth loss. Two different Swedish studies [7,8] observed that parity had an impact on tooth loss.

High parity negatively correlated with the number of remaining teeth. Conversely, Scheutz *et al.* [36] documented no link between parity and tooth loss in Tanzanian women, but they observed a relationship between parity and the principal cause of tooth loss—periodontal disease.

Thus, the association between parity and tooth loss remains unclear. There are complex confounding factors that are frequently not considered (including age, age at first birth, duration of birth intervals, breastfeeding duration, pre-existing nutritional status and food uncertainty) and this makes it difficult to draw conclusions regarding parity effects. It is also necessary to investigate the views of women on this issue to gain insights into their understanding and perception of parity and tooth loss. This is particularly critical in contexts with significantly differential parity and/or high parity. The perceptions of women are important as these may shape their behavior and attitudes towards oral health care. Women may resign themselves to a fate where pregnancy is expected to cause tooth loss, instead of seeking preventive care and dental treatment to maintain their oral health.

Hausa women of northern Nigeria were studied as part of a larger project on maternal and child oral health in a high parity population. Previously, the prevalence of caries experience and tooth loss in women from the region was found to be high. This was attributed to high consumption of sugars, low socio-economic status and poor utilization of oral health services [37]. However, the oral health conditions of the women may be related to their high parity. According to the latest available statistics, Nigeria has a high total fertility rate of 5.7 children per woman. In the northwest zone of the country where the Hausa live, it is 7.3 children [38]. This is ascribed to very early age at marriage, high levels of teenage childbearing, the male dominated social hierarchy, polygyny and low use of modern contraceptives [38,39].

## Methods

### Study design

A grounded theory approach, involving in-depth focus group discussions (FGDs) of high and low parity Hausa women in northern Nigeria, was implemented. This approach is suitable for obtaining data on married women's views regarding parity and tooth loss. A repetitive method of data collection and analysis was employed to develop a theoretical explanation of perceptions grounded in the data collected from the discussions with Hausa women.

### Sample population

The sample population was selected through a household survey in the Kumbotso Local Government Area (LGA) of Kano State, Nigeria using a multi-stage random sampling technique. Kano State is located in the northwest zone of Nigeria and has a population of 9.4 million [40]. Kumbotso LGA has its headquarters in the town of Kumbotso. The population is 295,979 people who live in an area of 158 km$^2$. The LGA consists of 11 administrative wards. According to the 2006 census, 66,010 women aged 15–65 years reside in Kumbotso LGA. Six wards were randomly selected from the LGA. Within each ward, two communities were randomly selected and all households in each community were approached.

### Group composition

A purposive selection [41] of women from different age cohorts and parity levels was identified from the participants in a general study on maternal and child oral health in Kano. The sample consisted of 33 women aged 19–66 years with the size determination based on the theoretical saturation concept of Grounded Theory [42]. Women of all parity levels were included.

Participants were grouped into three age cohorts (19–30 years, 31–45 years and 46–66 years) and each group consisted of an average of five women.

## Data collection method

Trained bilingual Hausa and English-speaking married women with previous experience in qualitative interviewing, along with the principal investigator, conducted the FGDs. The use of local Muslim Hausa women as field workers helped facilitate access to women in seclusion, promoted openness among the women during the FGDs, and minimized the potential objections and suspicions of participant's husbands. The local field workers were not assigned to groups in their own areas. Two FGD sessions were conducted per age cohort, for a total of six gatherings. Sessions were conducted in a quiet meeting room, and the discussions were moderated with the use of an interview guide that was prepared before the sessions.

The FGDs obtained responses to queries on the following topics: causes of tooth loss, effects of tooth loss on the quality of women's lives, issues regarding parity and tooth loss, and cultural beliefs on parity and tooth loss.

All interviews were taped, transcribed, and translated verbatim from Hausa into English. Two Hausa language teachers at Bayero University, Kano, and two Hausa-speaking dentists (also from Kano State and not involved in the study) supervised the transcription and translation.

## Ethical considerations

Ethical clearance for the study was obtained from the Ethics and Research Committee of Obafemi Awolowo University, Ile-Ife, Osun State, Nigeria (IPHOAU/12/717) and from the Human Research Ethics Committee of the University of the Witwatersrand, Johannesburg (M170343). A male local assistant who could speak Hausa fluently was employed to facilitate links with village leaders and husbands. Permission was obtained from local village leaders to conduct the study and informed consent was obtained from the husbands of married women living with their husbands. Written informed consent was obtained from each participant. The consent form was translated into Hausa and read to the participants who were not literate. Women who were not literate and were willing to participate in the study thumb printed on the consent form.

## Data analysis and theory building using the grounded theory approach

**Coding and theorizing.** Using the grounded theory approach, open, focused and axial coding was employed [42–47]. Throughout the analysis, memoing was done to facilitate the hypothesis formulation [48]. Two major theoretical categories were generated around the assumptions and beliefs attached to childbearing and tooth loss in women: the causes of tooth loss and the effects of tooth loss on women of childbearing age.

Network diagrams were drawn using the network view function of ATLAS.ti to show the relationships between categories and to display the models to explore the data and visualize the ideas and findings [49]. Illustrative direct quotations were drawn from the text to highlight key findings.

## Results

The mean age of the participants was $40.82 \pm 14.99$ years. More than half (54.5%) of the respondents had no formal education. Eighteen (54.5%) had five or more children. The mean parity was $4.48 \pm 3.00$. There was a significant difference between the mean parity levels by age

**Table 1. Socio-demographic characteristics of participants.**

| Characteristics | Total N = 33 |
|---|---|
| **Age (years)** | |
| 19–30 | 11 (33.3%) |
| 31–45 | 12 (36.4%) |
| 46–66 | 10 (30.3%) |
| **Level of education** | |
| Koranic only | 18 (54.5%) |
| Primary (partial or completed) | 5 (15.2%) |
| Secondary (partial or completed) | 10 (30.3%) |
| **Number of children (living)** | |
| ≤ 4 | 15 (45.5%) |
| 5–8 | 14 (42.4%) |
| > 8 | 4 (12.1%) |
| **Mean parity by age cohorts** | |
| 19–30 | 2.18 ± 1.60 |
| 31–45 | 6.25 ± 2.22 |
| 46–66 | 4.90 ± 3.51 |
| | p = 0.002 |
| Mean age (years) | 40.82 ± 14.99 |
| Mean parity | 4.48 ± 3.00 |
| **Occupation** | |
| Traders | 26 (78.8%) |
| Housewives | 7 (21.2%) |

group (p = 0.002). The majority of the women (78.8%) were small-scale informal traders in the market and the remainder (21.2%) were housewives (Table 1).

## Key findings and interpretation

The findings from this study reflect the beliefs and thoughts of Hausa women of child bearing and post childbearing ages. The themes included "Perception on the relationship between tooth loss and childbearing", "Beliefs about tooth loss", "Causes of tooth loss", and "Effects of tooth loss on women".

The network diagram [Fig 1] shows the perceived causes of tooth loss among Hausa women of childbearing and post-childbearing age. These etiologies include dirty mouth, vomiting during labor, excessive sugary diet, aging, food impaction, tooth worm infection and hole in the tooth, drinking cold water, road traffic accidents, other accidents, fights, extractions or loss due to poor tissue support and cancer. These causes are further described below.

## Array of reported causes of tooth loss

**Poor oral hygiene.** Respondents identified poor oral hygiene as a major cause of tooth loss among women. Because most women do not practice good dental care, they have tooth decay and tooth loss. Kolanut, a mild stimulant chewed to restore vitality and reduce hunger pangs, results in staining of teeth and was associated with poor oral hygiene.

> . . ..because of dirty mouth and kolanut chewing. Kolanut turn the colour of the teeth bad and make them dirty

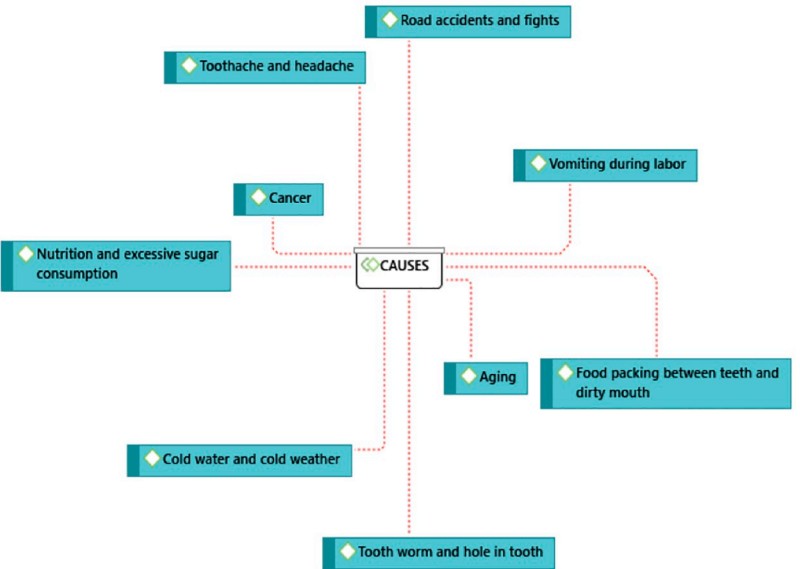

**Fig 1. Causes of tooth loss during childbirth.**

> . . ..*R3*, *REC 013*, *Hausa community*

> . . .. *Some women are unable to take care of their teeth because they do not care about the mouth*

> . . ..*R2*, *REC 010*, *Hausa community*

> . . ..*lack of brushing the teeth will make them fall out.*

> . . ..*R1*, *REC 012*, *Hausa community*

> . . . . . . ..*poor personal cleaning make the mouth to smell after sometime the teeth will be shaking and fall out.*

> . . ..*R5*, *REC 015*, *Hausa community*

**Food impaction, tooth worm infection and caries.** It was also discovered during the discussions that food impaction, inadequate dental care during pregnancy, and worms infesting the oral cavity lead to tooth loss. Worms create the holes seen in teeth, cause caries, make the teeth weak and eventually they are lost. Below are responses to support these points:

> . . ..*meat impaction between teeth.*

> . . ..*Hole made by worms*

> . . ..*R3&R5*, *REC 015*, *Hausa community*

> . . .*Tooth worms eat the tooth and make a hole in it. This cause tooth pain*

> . . .*Tooth worms make holes in teeth and cause pain*

*. . . I hear traditional medicine people say that tooth worm make holes in teeth and later gives pain and the tooth will be removed*

*. . . . . .dirty mouth have worms that damage the teeth*

*. . ...R1, R2, R3 & R4, REC 011, Hausa community*

**Aging.**   Discussants stated that women age faster than men due to childbearing and domestic chores. These cause them to lose their teeth earlier than men.

*. . ...Women become old earlier than men and it is because of childbearing. Because of this women lose more teeth than men*

*. . ...My friend who left the city for the village now has missing teeth maybe because she works more and she is not able to take care of her body. She is going to become old earlier than the people in the city.*

*. . ...R2 & R3 REC 012, Hausa community*

*. . ...as you grow old the teeth are no more strong, they are weak and will fall out one by one*

*. . .,R6, REC 011, Hausa community*

**Nutrition and excessive sugar consumption.**   Furthermore, it was mentioned during the course of the discussion that pregnant women enjoy eating excessive sugar and sugary foods. This leads to weak teeth, causing toothache and easy loss of the teeth. It was unanimously agreed that women mostly lose more teeth than men because of excessive sugar consumption.

*. . ...maybe she chew gums which contains sugar and can cause tooth hole and tooth loss.*

*. . ...R2, REC 012, Hausa community*

*. . ..consumption of sugar coating food make holes in teeth*

*. . ...consumption of honey, it is sweet and can cause holes in teeth*

*. . ...R2& R6, REC 013, Hausa community*

*. . ...girls like to eat sweets and chew gum, these make the teeth go bad. They cause holes and pain and the teeth are removed later*

*. . ...R1, REC 011, Hausa community*

**Cancer.**   Respondents suggested that women of childbearing age could lose their tooth if they have cancer of the mouth.

*. . ..Maybe the women that lost teeth have cancer*

*. . ... R6, REC 010, Hausa community*

*. . .. Cancer, big or small makes the teeth to be shaking and fall after some time*

. . . . . . *R1, REC 014, Hausa community*

**Other causes of tooth loss.**   Respondents offered that using the teeth to open bottled drinks could cause pain/trauma leading to loss of teeth. In addition, it was mentioned that teeth can be lost through fights or road traffic accidents and can also fall off by themselves (perhaps due to periodontal disease) or through extraction.

*. . . .. using teeth to open drink/ Coca-Cola crown tops or biting something hard. The tooth can break or fall out*

*. . . . . . "Okada" (motorcycle) or car accidents can make teeth fall out if you hit you mouth on the"Okada"or ground*

*. . . ..R2, REC 015*, **Hausa community**

*. . . . My tooth was removed because of pain. There was a hole in it.*

*. . . .Yes, my tooth came out on it own. It was shaking before it fell off*

*. . . .R2& R3, REC 010*, **Hausa community**

*. . .punch or slap on the face from fights between two people can make teeth fall off*

*. . .,R4, REC 013*, **Hausa community**

Furthermore, it was mentioned that ingestion of cold water and cold weather could lead to tooth loss.

*. . . . . . drinking cold water and cold weather. . . . . .*

*. . . ..R6, REC 011*, **Hausa community**

**Effects of losing teeth.**   Discussants described the effects of tooth loss in women of childbearing age as including esthetic decline, loss of self-esteem, eating and chewing difficulties, and financial burden.
Respondents said that tooth loss affects their self-esteem. Sometimes they lose their joy and are unable to speak in public and smile because of the shame of their lost tooth.

*Very painful associated with missing of joy because you cannot talk or laugh like you want to in public*

*. . . ..R2, REC 014*, **Hausa community**

*Esthetic problem [I can't laugh well because the gap (edentulous space) will show]*

*. . . ..R3, REC 010*, **Hausa community**

*R3: Esthetic/ loss of self-esteem [I don't like to laugh or talk in public so that people will not laugh at me]*

*R5: It affects the beauty [when you lose your front teeth you don't look beautiful again]*

*R6: Unable to smile*

*. . . ..R3*, *R5 &R6 REC 012*, *Hausa community*

In addition, tooth loss makes eating and chewing difficult for them. It was noted during the course of discussion that tooth loss causes pain and discomfort during mastication.

*. . . difficulty in eating/ chewing—I am unable to chew meat or maize, I miss out on them*

*. . . . . ..anytime I try to chew meat or maize, it is discomforting when the food is on the gum at the back where there are no teeth. I don't enjoy it.*

*. . . ..because it is front teeth I don't have difficulty in chewing but I don't feel pretty like I was when the teeth were there.*

*. . . ..R3&R5 REC 011*, *Hausa community*

Finally, respondents complained about the financial burden to replace their teeth when they lose them. They mentioned that it is expensive to replace missing teeth.

*. . . ..unable to chew and to replace with artificial teeth you have to pay money [Financial burden]*

*. . . ..R1*, *REC 011*, *Hausa community*

**Perceptions and beliefs on the relationship between parity and tooth loss.** Respondents were unanimous in their belief that there is a relationship between tooth loss and childbearing [Fig 2].

Respondents said that during the course of labor, women vomit, which makes their teeth weak and results in tooth loss. Excerpts from interactions with the respondents include:

*. . . ..Yes. I think because I know a woman with five children and she lost some teeth.*

*. . . ..R2*, *REC 015*, *Hausa community*

*. . . . . . .Vomiting during labor in female, which will make the teeth weak and will not last. Yes, I am having vomiting during labor and only one tooth lost after menopause.*

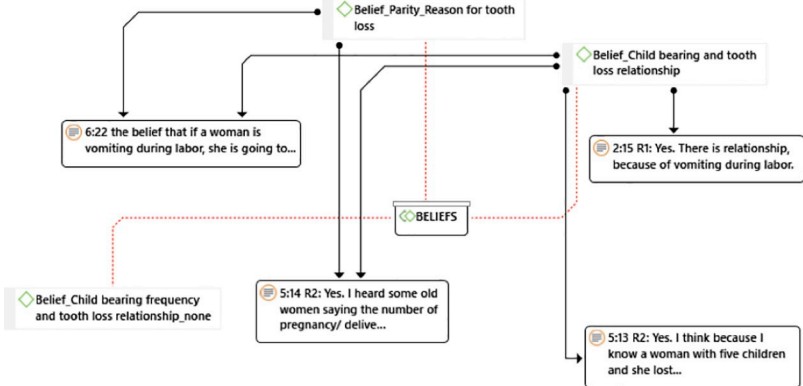

**Fig 2. Thematic beliefs of women on the relationship between parity and tooth loss.**

*. . ..R2*, *REC 010*, *Hausa community*

*. . ..Yes, I do vomit during labor and I lost my tooth [two teeth]*

*. . ..R3*, *REC 012*, *Hausa community*

*. . .We heard that when women are having vomiting during labor their teeth are not going to last.*

*. . ..R1*, *REC 013*, *Hausa community*

*.. I vomited during labor but did not lose a tooth but I am afraid that I will lose one soon or later*

*. . .. . .R4*, *REC 014*, *Hausa community*

*. . .. I know that pregnancy will cause lose of hair, nail, tooth. It also causes pain in tooth. When I was pregnant I lost my hair but not my tooth or nail.*

*. . .. . .R5*, *REC 011*, *Hausa community*

*. . .. . ..the number of pregnancy that a woman has is the number of teeth she will lose.*

*. . .. . .R6*, *REC 016*, *Hausa community*

*. . .. . ... I lost a tooth because I vomited when I was in labor*

*. . .,R7*, *REC 015*, *Hausa community*

During the course of the discussion, respondent gave a local phrase "***payar baka***" for vomiting during labor. Quotes to buttress this are:

*. . .. . . Vomiting during labor [payar baka]*

*. . ..R1*, *REC 010*, *Hausa community*

*. . ... I vomited when I was going to have my last child and my tooth fell off*

*. . ..R3*, *REC 013*, *Hausa community*

However, some respondents believed there is no relationship between tooth loss and child bearing. Excerpts from interaction with the respondents are shown below:

*. . ... (Q: Any relationship between parity and tooth loss?). It's not due to pregnancy. It's due to guava seed impaction between the teeth but that happened during pregnancy, I have 12 children without losing any tooth.*

*. . ..R1*, *REC 012*, *Hausa community*

*. . .. . .. I lost some teeth before I marry so it cannot be pregnancy that cause it since I was not pregnant before I marry*

*. . ..R3*, *REC 014*, *Hausa community*

*. . . . . . No (Q7: Is there any relationship between the number of children women have and the number of teeth lost?)*

*. . . .. No [unanimously]*

*. . . ..R1&6, REC 011, Hausa community*

Some respondents stated that tooth loss is not associated with pregnancy or childbirth. It emerged that this belief was based on the observations that young girls can also have tooth decay that leads to tooth loss. Excerpts from interactions with the respondents are provided below:

*. . . . . .some women also have holes in teeth associated with tooth loss but not really associated with childbearing. Because even a very young girl can have holes in teeth associated tooth lost.*

*. . . . . ..I lost some teeth before I marry or even have children. It cannot be pregnancy maybe the sweets and chewing gum I took when I was young.*

*. . . ..R6 & R7, REC 010, Hausa community*

## Discussion

To the best of our knowledge, this is the first qualitative study to explore the perceptions of women on parity and tooth loss. Our research provides an in-depth understanding of women's perceptions regarding the issue in a society where high parity is the norm. All the discussions were conducted in the participant's local language (Hausa) and environment. This was to reduce misinterpretation, possible bias and false response [50]. The separation of participants into different age groups allowed for diverse experiences and perceptions within the cultural group to be expressed.

Generally, the women in this study did not perceive that the number of children is directly associated with tooth loss. Pregnancy was indirectly related to tooth loss via the belief that a woman is prone to lose a tooth or at risk of losing one in future if she vomits during childbirth—the condition known as "payar baka". Those who vomited during childbirth but did not lose a tooth were afraid that they would lose a tooth or more teeth later in life. The participants could not explain the basis for the link between vomiting during labor and tooth loss. Women could be conflating forceful vomiting with expelling of teeth that are lost due to weakened gingival and periodontal tissues.

Vomiting, the forceful expulsion of gastrointestinal content through the mouth, is common during labor. It is often a result of anxiety and nervousness about the labor process. In addition, severe labor pains and contractions can induce vomiting. A literature search on vomiting during labor and tooth loss did not yield any further information. It is possible that the belief of the Hausa women is specific to their population.

The participants in this study offered various causes for tooth loss. One of these was "dirty mouth". Dirty mouth, often the result of inadequate or no tooth brushing, is characterized by accumulation of plaque and calculus deposits. These deposits harbor microorganisms that produce toxins, which are harmful to the supporting structures of the teeth. This situation leads to gingivitis, and if untreated may progress to periodontitis and possibly tooth loss. Pregnancy is associated with gingivitis and periodontitis, with progression of these oral conditions during the pregnancy [51]. In addition, women with previous pregnancies have higher gingival index scores and periodontal probing depths than those pregnant for the first time [52–54]. It is therefore likely that pregnancy has an indirect effect on the supporting structures of the teeth

that may affect tooth loss. However, the women in the focus groups did not mention the acute symptoms associated with pregnancy gingivitis.

Worm infection is another factor our respondents believed to cause tooth loss. Tooth worm was a widely held concept historically. There are oral and written accounts of tooth worms and tooth decay in antiquity [55]. The worms were believed to burrow through the teeth to cause decay and pain. Similarly, tooth worms were linked to periodontal disease. In Africa, guinea worms from infected drinking water were considered to be a cause of tooth decay because the worms breed in cool water settings such as deep wells [56]. Presently, deep cool wells serve as major sources of drinking water in developing countries but not in developed countries [57]. Consumption of rotten, maggoty cheese was another suggested source of tooth decay [58]. The Sudanese believed in the concept of tooth worms ('sosa') from infested foods as a cause of tooth decay [59]. The tooth worm theory of tooth decay was first discredited by Fauchard (1678–1761) and further disproved following the Age of Enlightenment [60]. For the Hausa people, the tooth worm concept may persist because worm infestation is a reality of their daily lives. Guinea worms, now approaching eradication, were until recently endemic in regions of northern Nigeria [61]. Their breeding in cold water may tie into Hausa women's belief that drinking cold water is a cause of tooth loss.

Some of the respondents in our study perceived that sugary foods, including chewing gums, are associated with tooth loss from carious decay. The origin of this belief is most likely a reflection of Hausa women's exposure to modern dentistry and health awareness. Even the packaging of common products, such as chewing gums, frequently includes messages about the effects of sugar. The Hausa women in our study related sugar consumption to pregnancy cravings. This may be culturally specific or dependent on local food resources, as pregnancy cravings for non-sugary foods in Africa are known to include clay and soils [62,63]. While most studies document higher caries rates in women than men [28,30–33], differential consumption of sweet foods has not been directly implicated as the cariogenic agent. Walker and Hewlett [28] attributed the heightened caries experience of women to their involvement in food processing and preparation, and with snacking patterns that enhance the cariogenic oral environment. Pregnancy and parity are related to caries experience in some groups, with women of high parity having more caries in studies of Thai and US women [64,65]. However, there was no link between parity and caries in women from Tanzania and black South African women [4,36]. This is an area requiring further research with more detailed examination of dietary variability and oral health practices amongst women of high parity.

The participants in this study believed that women age faster than men due to the stress of reproduction and household chores. Consequently, women lose more teeth than men. The natural process of ageing is universally linked with tooth loss. Periodontal disease, caries and tooth wear become worse and predispose an individual to tooth loss as they age. Maintaining good oral hygiene can be difficult in old age due to poor mobility or the inability to carry out fine movements for effective plaque removal [66]. In addition, ageing comes with health challenges such as diabetes, cardiovascular disease and the use of medications that can cause gingival swelling and xerostomia (dry mouth) [66]. These health challenges, if not adequately managed, further deteriorate the oral health leading to tooth loss. High parity women are prime candidates for oral health complications. They are at greater risk of diabetes and cardiovascular disease [2]; a systematic review concluded that adults with diabetes have an increased risk of onset and progression of periodontal disease [67]. At present, it is unknown if the advancement of aging described by Hausa women refers to their general strength and energy or if it includes some of these other age-related issues.

Tooth loss impacts greatly on the quality of life, and the participants in this study perceived that the loss of one or more teeth affects mastication, their appearance, their sense of well-

being and their financial status. Functional and esthetic impairments are widely reported in the literature as major effects of tooth loss on individual quality of life [11,14,68]. As tooth loss is associated with ageing, individuals may perceive that they are old following the loss of a tooth. Affordability of dental care may be a challenge, especially in developing countries. The participants in this study perceived that replacement of a missing tooth was expensive. This may further reduce their quality of life because their hope of tooth replacement is low.

Hausa women's views on tooth loss illustrate that their knowledge base incorporates traditional beliefs (*payar baka*, tooth worm, sugar cravings), cultural notions of aging and attractiveness, and Western dental concepts regarding oral hygiene. These findings have clear implications for future health initiatives, which need to address the complexity of existing beliefs when working to change oral health behaviors.

The results of this study cannot be generalized because they represent the perspectives of a specific group of participants. It is possible that a different group of participants may have different views. However, this limitation was taken into consideration by involving women of different age groups and parity levels.

## Conclusion

The study participants offered various reasons for tooth loss, including poor oral hygiene, vomiting during childbirth, tooth worm and ageing. The respondents perceived gender inequality in tooth loss, with a tendency towards more loss in females. Masticatory and esthetic impairments were the most perceived negative effects of tooth loss on the quality of life. Generally, the women in this study do not perceive parity per se as directly associated with tooth loss. They associated childbirth experiences, particularly vomiting, as the cause of tooth loss. The fact that the condition was identified with a specific term in the Hausa language ('*payar baka*') illustrates that it describes a common occurrence with relevance to women's health knowledge and oral health behaviors.

## Supporting information

**S1 Data. Bio data.**
(XLSX)

**S2 Data. Questionnaire data.**
(DOC)

## Acknowledgments

We thank the women who participated in the discussions and shared their experiences and knowledge. We also thank the field team who helped with establishing the project, provided local support in the Kano region, and translated the transcripts from Hausa to English.

## Author Contributions

**Conceptualization:** Elizabeth O. Oziegbe, Lynne A. Schepartz.

**Data curation:** Elizabeth O. Oziegbe.

**Formal analysis:** Elizabeth O. Oziegbe.

**Funding acquisition:** Lynne A. Schepartz.

**Investigation:** Elizabeth O. Oziegbe.

**Project administration:** Elizabeth O. Oziegbe.

**Supervision:** Lynne A. Schepartz.

**Writing – original draft:** Elizabeth O. Oziegbe.

**Writing – review & editing:** Lynne A. Schepartz.

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
