## [Decision Letter · Decision Letter 0]

10 Sep 2019

PONE-D-19-21577

Is parity a cause of tooth loss? Perceptions of northern Nigerian Hausa women

PLOS ONE

Dear Dr. Schepartz,

Thank you for submitting your manuscript to PLOS ONE. After careful consideration, we feel that it has merit but does not fully meet PLOS ONE’s publication criteria as it currently stands. Therefore, we invite you to submit a revised version of the manuscript that addresses the points raised during the review process.

We would appreciate receiving your revised manuscript by Oct 25 2019 11:59PM. To enhance the reproducibility of your results, we recommend that if applicable you deposit your laboratory protocols in protocols.io, where a protocol can be assigned its own identifier (DOI) such that it can be cited independently in the future. For instructions see: http://journals.plos.org/plosone/s/submission-guidelines#loc-laboratory-protocols

We look forward to receiving your revised manuscript.

Kind regards,

Denis Bourgeois

Academic Editor

PLOS ONE

Journal Requirements:

Reviewers' comments:

Reviewer's Responses to Questions

**Comments to the Author**

1. Is the manuscript technically sound, and do the data support the conclusions?

Reviewer #1: Yes

Reviewer #2: Partly

2. Has the statistical analysis been performed appropriately and rigorously? 

Reviewer #1: Yes

Reviewer #2: Yes

3. Have the authors made all data underlying the findings in their manuscript fully available?

Reviewer #1: No

Reviewer #2: Yes

4. Is the manuscript presented in an intelligible fashion and written in standard English?

Reviewer #1: No

Reviewer #2: Yes

5. Review Comments to the Author

Reviewer #1: The paper concerns an original issue with interesting data on subjective perception in a sample of women in a developing country about the relationship between tooth loss and parity.

Statistical and methodological procedures are described and conducted with the Focus Group Discussion, which is a correct way to assess subjective perceptions in specific cultural environments.

Besides, there are several redundancies within the whole text that require an optimization of the text itself that can lead to better overall readability; a thorough English language review is needed as well, in particular in the abstract, introduction, tooth loss and discussion sections.

Moreover, authors should explain what the Kolanut is (mentioned in the "Poor Oral Hygiene" paragraph page 9) and should correct the concept stating that untreated gingivitis leads authomatically to periodontitis reported at page 15, second paragraph, line 5. Author should express the possibility of periodontitis development, and not its certainty.

Reviewer #2: • The introduction is too long and dispersed. Refocus on the dental problem

• "Parity has been linked to tooth loss, although the association remains poorly investigated".

This sentence is difficult to understand. Please clarify the meaning

• "Tooth loss impacts greatly on the quality of life. The participants in this study perceived that the loss of one or more teeth affects mastication, their appearance, their sense of well-being and their financial status. Functional and esthetic impairments are widely reported in the literature as major effects of tooth loss on individual quality of life [20,23,77]. Fiske et al. [23], in a qualitative study of edentulous patients, documented bereavement, loss of self-confidence, concerns about appearance and self-image as part of the emotional responses to tooth loss. As tooth loss is associated with ageing, individuals may perceive that they are old following the loss of a tooth. Affordability of dental care may be a challenge, especially in developing countries. The participants in this study perceived that replacement of a missing tooth was expensive. This may further reduce their quality of life because their hope of tooth replacement is poor".

This sentences are not one of the objectives of the study.

• Dental references must be updated

• Why do not the authors speak about pregnancy gingivitis, a classic clinical sign between the 3rd month of pregnancy and 45th after childbirth?

Comments for the editor:

Original qualitative study but whose main criticisms reside in obsolete references associated with a need to condense the introduction

6. PLOS authors have the option to publish the peer review history of their article (what does this mean?). If published, this will include your full peer review and any attached files.

Reviewer #1: No

Reviewer #2: No

---

## [Author Response · Author response to Decision Letter 0]

16 Oct 2019

Response to Reviewers

Note: Due to revisions and deletions, the locations of deletions (Page #, paragraph and line) do not pertain to the new text.

A thorough English language revision of the entire text was done to improve the flow in certain sections.

Reviewers 1

1. There are several redundancies within the whole text that require an optimization of the text itself that can lead to better overall readability; a thorough English language review is needed as well, in particular in the abstract, introduction, tooth loss and discussion sections.

We have deleted the following:

Introduction

Page 3 paragraph 2 deleted

High parity, when a woman has given birth five or more times regardless of whether the child survived, (also referred to as grand multiparity [7]) adversely affects maternal morbidity and in some cases contributes to mortality [8-10]. Increased number of children is associated with greater risk of obesity, diabetes and cardiovascular disease, especially in developed countries [11-14]. Conversely, high parity is linked to lower maternal morbidity in developing countries [15]. In the latter environments, healthier women may have more children simply because they have better access to resources that can sustain pregnancy.

Tooth loss

Page 3 paragraph 1 lines 1-2 – deleted

Teeth play important biological and aesthetic roles in the life of an individual. The loss of a tooth is associated with negative social, functional and psychological consequences.

Page 3 paragraph 1 last line –deleted

The loss of teeth has negative consequences on the individual irrespective of the reasons for the loss or the life stage in which it occurs. 

Page 4 paragraph 3 lines 4-5 –deleted

Important factors include attitudes toward dental care and availability of resources for treatment.

Page 4 last paragraph line 2 – deleted

While most researchers used quantitative data to establish a link between the two,

Page 5 paragraph 1 lines 4-5 – deleted

Therefore, the purpose of this study was to understand the beliefs among Hausa women through development of a theoretical framework explaining the pathways linking their views on parity and tooth loss. 

Discussion 

Page 15 paragraph 2 lines 1-3 – deleted

Parity has been linked to tooth loss, although the association remains poorly investigated. This study provides information on parity and tooth loss beliefs in women from northern Nigeria.

Page 15 paragraph 2 line 6 – deleted

Some of the women in the study experienced the phenomenon while

Page 18 paragraph 2 line 5– 7 - deleted

Fiske et al. [14], in a qualitative study of edentulous patients, documented bereavement, loss of self-confidence, concerns about appearance and self-image as part of the emotional responses to tooth loss

2. Authors should explain what the Kolanut is (mentioned in the "Poor Oral Hygiene" paragraph page 9) and should correct the concept stating that untreated gingivitis leads authomatically to periodontitis reported at page 15, second paragraph, line 5.

Kolanut has been explained under “Poor Oral Hygiene” page 10 paragraph 1.

Kolanut, a mild stimulant chewed to restore vitality and reduce hunger pangs, results in staining of teeth and was associated with poor oral hygiene. 

Re untreated gingivitis. Thank you. Agreed. The concept that untreated gingivitis automatically leads to periodontitis has been amended. We expressed the possibility of periodontitis and not its certainty – Page 16 paragraph 1 

Response to Reviewer 2

3. The introduction is too long and dispersed. Refocus on the dental problem

We made deletions in response to this comment and to Reviewer 1. 

See point one for Reviewer 1 above.

4. "Parity has been linked to tooth loss, although the association remains poorly investigated".

This sentence is difficult to understand. Please clarify the meaning

Thank you. The sentence is deleted as unnecessary as the point is made elsewhere. 

5. "Tooth loss impacts greatly on the quality of life. The participants in this study perceived that the loss of one or more teeth affects mastication, their appearance, their sense of well-being and their financial status. Functional and esthetic impairments are widely reported in the literature as major effects of tooth loss on individual quality of life [20,23,77]. Fiske et al. [23], in a qualitative study of edentulous patients, documented bereavement, loss of self-confidence, concerns about appearance and self-image as part of the emotional responses to tooth loss. As tooth loss is associated with ageing, individuals may perceive that they are old following the loss of a tooth. Affordability of dental care may be a challenge, especially in developing countries. The participants in this study perceived that replacement of a missing tooth was expensive. This may further reduce their quality of life because their hope of tooth replacement is poor".

This sentences are not one of the objectives of the study.

In the methods section, under data collection method, we referred to obtaining information on the effects of tooth loss on the quality of lives of the women. Page 6 paragraph 4. 

Thus the above should not be deleted because it is the aspect of the discussion on the participant’s responses to the effects of tooth loss on their quality of life.

We did delete the sentence on Fiske’s work as it was previously mentioned.

6. Dental references must be updated.

We are unable to respond directly to this comment, as it does not tell us specifically what areas need to be updated. We checked our references with this comment in mind. We do include a few ‘historic’ references where appropriate to provide context, but otherwise our cited literature is current. The sources on oral health for Northern Nigeria are, sadly, the latest available statistics.

7. Why do not the authors speak about pregnancy gingivitis, a classic clinical sign between the 3rd month of pregnancy and 45th after childbirth?

Pregnancy gingivitis, and the clear acute clinical symptoms associated with it, did not emerge as a cause of tooth loss in the focus group discussions. However, we did refer to it in our discussion under

---

## [Decision Letter · Decision Letter 1]

21 Nov 2019

Is parity a cause of tooth loss? Perceptions of northern Nigerian Hausa women

PONE-D-19-21577R1

Dear Dr. Schepartz,

We are pleased to inform you that your manuscript has been judged scientifically suitable for publication and will be formally accepted for publication once it complies with all outstanding technical requirements.

With kind regards,

Denis Bourgeois

Academic Editor

PLOS ONE

Additional Editor Comments (optional):

Reviewers' comments:

Reviewer's Responses to Questions

**Comments to the Author**

1. If the authors have adequately addressed your comments raised in a previous round of review and you feel that this manuscript is now acceptable for publication, you may indicate that here to bypass the “Comments to the Author” section, enter your conflict of interest statement in the “Confidential to Editor” section, and submit your "Accept" recommendation.

Reviewer #1: All comments have been addressed

Reviewer #2: All comments have been addressed

2. Is the manuscript technically sound, and do the data support the conclusions?

Reviewer #1: (No Response)

Reviewer #2: Yes

3. Has the statistical analysis been performed appropriately and rigorously? 

Reviewer #1: (No Response)

Reviewer #2: Yes

4. Have the authors made all data underlying the findings in their manuscript fully available?

Reviewer #1: (No Response)

Reviewer #2: Yes

5. Is the manuscript presented in an intelligible fashion and written in standard English?

Reviewer #1: (No Response)

Reviewer #2: Yes

6. Review Comments to the Author

Reviewer #1: (No Response)

Reviewer #2: (No Response)

7. PLOS authors have the option to publish the peer review history of their article (what does this mean?). If published, this will include your full peer review and any attached files.

Reviewer #1: No

Reviewer #2: No

---

## [Editor Report · Acceptance letter]

26 Nov 2019

PONE-D-19-21577R1 

Is parity a cause of tooth loss? Perceptions of northern Nigerian Hausa women 

Dear Dr. Schepartz:

I am pleased to inform you that your manuscript has been deemed suitable for publication in PLOS ONE. Congratulations! Your manuscript is now with our production department. 

With kind regards,

on behalf of

Professor Denis Bourgeois 

Academic Editor

PLOS ONE